# Design and Evaluation of an Automated Ultraviolet-C Irradiation System for Maize Seed Disinfection and Monitoring

**DOI:** 10.3390/s25196070

**Published:** 2025-10-02

**Authors:** Mario Rojas, Claudia Hernández-Aguilar, Juana Isabel Méndez, David Balderas-Silva, Arturo Domínguez-Pacheco, Pedro Ponce

**Affiliations:** 1Institute of Advanced Materials for Sustainable Manufacturing, Tecnologico de Monterrey, Monterrey 64700, Mexico; mario.rojas@tec.mx (M.R.); pedro.ponce@tec.mx (P.P.); 2Postgraduate in Sytems Engineering-Biophysical Sytems, Instituto Politecnico Nacional, Mexico City 07738, Mexico; clhernandeza@ipn.mx (C.H.-A.); fdominguezp@ipn.mx (A.D.-P.); 3School of Architecture, Art and Design, Tecnologico de Monterrey, Mexico City 14380, Mexico; isabelmendez@tec.mx

**Keywords:** UV-C system, seed disinfection, automation, seed treatment technology, design

## Abstract

This study presents the development and evaluation of an automated ultraviolet-C irradiation system for maize seed treatment, emphasizing disinfection performance, environmental control, and vision-based monitoring. The system features dual 8-watt ultraviolet-C lamps, sensors for temperature and humidity, and an air extraction unit to regulate the microclimate of the chamber. Without air extraction, radiation stabilized within one minute, with internal temperatures increasing by 5.1 °C and humidity decreasing by 13.26% over 10 min. When activated, the extractor reduced heat build-up by 1.4 °C, minimized humidity fluctuations (4.6%), and removed odors, although it also attenuated the intensity of ultraviolet-C by up to 19.59%. A 10 min ultraviolet-C treatment significantly reduced the fungal infestation in maize seeds by 23.5–26.25% under both extraction conditions. Thermal imaging confirmed localized heating on seed surfaces, which stressed the importance of temperature regulation during exposure. Notable color changes (ΔE>2.3) in treated seeds suggested radiation-induced pigment degradation. Ultraviolet-C intensity mapping revealed spatial non-uniformity, with measurements limited to a central axis, indicating the need for comprehensive spatial analysis. The integrated computer vision system successfully detected seed contours and color changes under high-contrast conditions, but underperformed under low-light or uneven illumination. These limitations highlight the need for improved image processing and consistent lighting to ensure accurate monitoring. Overall, the chamber shows strong potential as a non-chemical seed disinfection tool. Future research will focus on improving radiation uniformity, assessing effects on germination and plant growth, and advancing system calibration, safety mechanisms, and remote control capabilities.

## 1. Introduction

According to the United Nations Sustainable Development Goals (SDGs), particularly SDG 2, achieving food security requires productive, resource-efficient, and environmentally responsible agricultural practices [1]. Rising population pressure and environmental degradation demand higher yields without increasing ecological harm [2]. Although chemical pesticides are widely used to control crop diseases, their overuse threatens human health, ecosystems, and accelerates resistant pest strains [3], underscoring the need for sustainable alternatives. Maize (*Zea mays* L.), a staple crop essential to global food supply chains and livestock production [4], often suffers reduced seed viability during storage and planting due to oxidative stress, leading to poor germination and crop establishment. Innovative, non-destructive strategies are therefore required to enhance seed quality and resilience under diverse environmental conditions.

One promising strategy is seed priming, a physiological technique that prepares seeds to better withstand stress by conditioning them before germination using biological, chemical, or physical agents [5,6]. Seed priming has been shown to enhance germination rates, uniformity, seedling vigor, and resilience to both abiotic and biotic stressors [7,8]. As a low-cost, low-risk intervention, it aligns with sustainable agriculture goals by improving productivity while minimizing environmental impact.

Among physical priming methods, ultraviolet (UV) radiation, particularly in the UV-C range (200–280 nm), has emerged as a safe, eco-friendly, and cost-effective treatment [9]. Similar to other bioactive radiation types, UV radiation is widely recognized in pest and pathogen control strategies [10]. UV-C radiation exhibits potent germicidal effects, disrupting the DNA of pathogens and stimulating plant defense responses by inducing protective compounds such as terpenes and activating key stress–response pathways [11,12]. Controlled UV-C exposure has been shown to enhance crop productivity, yield quality, and resilience to environmental stress [8,13]. Positive effects have been reported for wheat, beans, maize, and oats when the dose is carefully optimized [14,15,16]. However, excessive irradiation can trigger oxidative stress, reducing seed viability [17].

Specialized UV-C irradiation chambers have been developed in various configurations. Stationary box-type systems and rotating cylinders fitted with low-pressure mercury lamps are common, with lamps typically placed above and below the seed bed to ensure uniform exposure. Tunnel-type units employ opposing lamps over a motorized conveyor, providing consistent dosing during seed transport Ali et al. [18]. Box-type chambers often line the interior walls with UV lamps; for example, El-Beltagi et al. [19] treated cauliflower seeds with UV-A (365 nm) and UV-C (254 nm) tubes mounted in an aluminum enclosure. Rotational geometries are also widespread: Hernandez-Aguilar et al. [20] installed four 254 nm lamps in a stainless-steel cylinder containing a rotating mesh drum to equalize doses for bean and lentil seeds Hernandez-Aguilar et al. [20,21]. Other examples include a compact 25 W aluminum cylinder for wheat seeds Mahmood and Ibrahim [14], a twelve-lamp 3D array for maize disinfection Popović et al. [15], parabolic reflectors for tomato priming Scott et al. [22], and combined UV-B/UV-C sources for oat enhancement Stefanello et al. [16]. These chambers improve germination or reduce microbial load but almost all rely on fixed exposure settings and lack in situ sensing of UV dose, temperature, or humidity, limitations that motivate the fully automated sensor-integrated system presented in this study.

At the same time, computer vision has gained prominence as a non-destructive, rapid, and reliable tool for maize seed analysis. Early work demonstrated the potential of machine vision and pattern recognition by extracting geometric, shape, and color features for seed variety identification with high accuracy [23]. More recent studies have integrated deep learning with feature extraction, such as Javanmardi et al. [24], who combined CNN-derived and handcrafted features to achieve over 98% accuracy in maize seed classification. Advances in deep neural architectures further improved performance: Li et al. [25] enhanced ResNet50 with attention mechanisms, while Xu et al. [26] employed an improved P-ResNet with transfer learning, achieving up to 99.7% accuracy. Beyond classification, computer vision has been applied to phenotyping and quality traits, with Yu et al. [27] using an optimized YOLOv5 [28] for automated 100-kernel weight measurement and Li et al. [29] developing a system for detecting multiple types of kernel damage. These advances highlight the potential of camera-based systems in intelligent agriculture and seed quality control.

Despite the documented benefits of UV-C seed priming, most existing chambers rely on fixed exposure settings and lack the real-time monitoring of UV dose, temperature, or humidity, limiting reliability, reproducibility, and scalability. Similarly, although computer vision has proven to be effective for seed analysis—accurately identifying varieties, phenotyping, and detecting damage—its integration with UV-C systems remains largely unexplored, preventing the real-time evaluation of treatment effects and adaptive optimization.

To address these gaps, this study introduces a fully automated UV-C priming chamber for maize seeds that integrates environmental sensors, irradiation monitoring, automated exposure control, and a camera-based computer vision module for post-treatment seed assessment. The system, built with two low-cost 8 W UV-C lamps, a modular coroplast lamp support, and a metallic enclosure, ensures consistent, safe, and replicable priming while remaining portable and accessible to researchers and smallholders. Prototype validation included experiments on germination rate, uniformity, exposure homogeneity, and environmental stability, providing insights for scalable, intelligent, and reproducible UV-C seed priming.

The integration of a camera to monitor maize seed color during irradiation is particularly innovative. Unlike prior computer vision applications in static post-harvest analyses, the camera continuously captures dynamic color changes as seeds undergo UV-C treatment, providing a non-destructive, quantitative indicator of priming effectiveness. Combined with real-time environmental sensing, this approach enables feedback-driven evaluation and optimization of exposure, transforming UV-C seed priming into an intelligent, adaptive process and extending the scope of computer vision in sustainable agriculture.

## 2. Materials and Methods

### 2.1. Prototype Development Methodology

The design process followed a structured methodology as illustrated in the workflow shown in Figure 1. Based on the approach described by Cogan [30], the process begins with identifying customer needs, followed by defining system requirements, developing the solution, and ultimately moving to manufacturing and testing.

### 2.2. Design Framework for Determining Prototype Specifications from Previous Work

Developing robust and scalable UV-C seed treatment chambers requires a systematic, multicriteria design methodology rather than ad hoc prototyping. The earliest comprehensive effort was undertaken by Méndez et al. [31], who applied the Penta-S model (Social, Sustainable, Sensing, Smart, Safe) [32] in combination with Fuzzy-TOPSIS, Quality Function Deployment (QFD), and a fuzzy morphological matrix to benchmark five Instituto Politécnico Nacional (IPN) prototypes. Their evaluation—ranging from a 3.5 W portable pouch (Bright Solution STERI BAG; Figure 2a) and a classroom desktop LED box (Figure 2b), to a 5 W SBS sterilizer with Qi charging (Figure 2c), a 36 W Sterilite tote for mid-volume batches (Figure 2d), and a 131 × 29 cm stainless-steel cylinder with a rotating mesh drum for industrial throughput (Figure 2e; [20])—showed that none offered closed-loop automation or real-time sensing.

As a result, Méndez et al. [31] proposed a prototype for future improvement, in which QFD translates stakeholder and agronomic needs (e.g., uniform dose and low-power consumption) into measurable engineering targets [33]. The morphological matrix was then used to evaluate subsystem options such as lamp type, sensors, airflow, and enclosure material [34], while TOPSIS or its fuzzy variant ranked the resulting configurations by performance, cost, and sustainability [35]. Through this process, the authors synthesized four new designs and identified the ”High Tech Design”—a UV-C chamber with an integrated sensor-rich control bay—as the optimal solution. Although still theoretical, this design addresses the major shortcomings of earlier prototypes by incorporating real-time UV dose monitoring, environmental sensing, automated timing, and enhanced user safety.

Consequently, the literature identifies three capabilities that are presently lacking in published chamber designs: (i) the implementation of closed-loop UV-dose control, (ii) the integration of real-time micro-climate sensing mechanisms, and (iii) the development of a lightweight, cost-effective structure that allows replication using commercially available components. Addressing these high-level deficiencies to generate a viable prototype necessitates a rigorous process of requirement capture and option screening.

### 2.3. Requirements Definition and Morphological Analysis

According to agricultural researchers at the IPN and their previous work with earlier prototypes (Figure 2), the primary application of the proposed UV-C irradiation system is for microgreens—such as maize, spinach, kale, arugula, chia, lentil, and basil. In many greenhouses, microgreen production typically requires 20–25 g of seed per variety, with growers often cultivating 2 to 10 different varieties per week. Within this context, maize seed is an important component, and the chamber’s capacity and modular design are well-suited to these weekly operational demands. The prototype’s scale ensures functionality for continuous, small-batch seed conditioning, aligning with the production cycles of microgreen growers. Of course, its frequency of use will ultimately depend on the specific market demand for microgreens, but the system provides the flexibility and throughput needed for consistent, week-to-week operation.

Alongside the automated system development, agricultural researchers required a computer vision module. Its primary purpose is to demonstrate the feasibility of integrating image-based monitoring without requiring the immediate development of a fully customized training dataset for every design modification. By capturing and analyzing color changes during the irradiation process, this module provides a flexible foundation for the future integration of advanced image analysis and feedback-driven control, ultimately paving the way toward intelligent, adaptive UV-C seed priming systems.

Building on the recent theoretical framework [31], the technical requirements for the prototype were defined across seven functional domains: (1) Disinfection Function, (2) Environmental Monitoring and Process Control, (3) Interface and Data Management, (4) Structural and Mechanical Design, (5) Safety and User Feedback, (6) Sustainability and Community Integration, and (7) Computer Vision and Smart Analysis. To translate these requirements into concrete design options, a morphological matrix (Table 1) was developed to systematically explore alternatives for each function [31,36]. This method decomposed the chamber into subsystems—such as disinfection sources, enclosure materials, sensing modules, and user interfaces—and organized multiple technological choices for each. By structuring the design space, the matrix broadened innovation pathways while enabling transparent trade-off analysis. The resulting configurations were then evaluated against criteria such as cost, development time, reliability, scalability, and regulatory compliance, ensuring that the final design balanced technical feasibility with practical deployment needs.

### 2.4. System Architecture of the Automated UV-C Seed Disinfection Chamber

The selected solution represents an optimal combination of alternatives summarized in Table 2. The alternatives represent the final prioritized solutions, which were consistently rated as the most convenient by the evaluation group. By integrating the technical requirements with the prioritized design options identified through the morphological matrix, the prototype addresses stakeholder needs systematically rather than through ad hoc solutions. As a result, the prototype emerges as a validated configuration that assures functional reliability, environmental stability, and practical feasibility for researchers and smallholders.

Figure 3 presents the overall architecture of the automated UV-C irradiation system developed for maize seed disinfection. The diagram shows the integration of the electronic control system, the user-interaction modules, and the disinfection chamber. At its core, a microcontroller governs the operation of two UV-C lamps, the auxiliary lighting, and the air extraction system, while simultaneously acquiring real-time data from environmental sensors. This information is communicated through serial connection to a Raspberry Pi 4, which hosts a graphical user interface (GUI) to monitor sensor readings and configure exposure duration.

The chamber consists of a metallic enclosure with a modular lamp support, ensuring portability, radiation containment, and structural stability. An internal camera captures images of seeds during treatment, which are analyzed on a separate computer through the vision system interface to extract color metrics as indicators of treatment effects. This computer vision module demonstrates the feasibility of image-based monitoring during irradiation and establishes a foundation for the future integration of advanced analysis and feedback-driven control toward intelligent, adaptive UV-C priming systems. A detailed description of these components is provided in the following subsections.

#### 2.4.1. Designed Lamp Support Structure for UV-C Irradiation

The chamber features a custom-designed internal support structure to securely mount two 8W UV-C lamps in a horizontal face-to-face configuration. Located centrally between the lamps is an adjustable tray holder that accommodates the seed tray during the disinfection process. This adjustability allows for precise control of the distance between the lamps and the tray, allowing optimization of UV-C radiation intensity according to the required exposure level.

The designed structure was built using 4 mm coroplast, a lightweight and durable corrugated plastic material. Coroplast was chosen for its ease of cutting and assembly, making it an ideal choice to quickly construct a functional and adaptable prototype as shown in Figure 4.

#### 2.4.2. Electronic Control and Monitoring System

The core for controlling the timing of UV-C radiation exposure and data acquisition is an Arduino Nano 33 BLE development board (Arduino, Turin, Italy), which integrates a Nordic Semiconductors nRF52840 microcontroller (Nordic Semiconductor ASA, Oslo, Norway). This microcontroller features a 32-bit ARM® Cortex® M4 architecture that operates at 64 MHz. The development board includes several digital input/output ports, analog inputs, and serial and I2C communication interfaces. Digital output pins are used to trigger the activation of the two 8 W UV-C lamps, an auxiliary bulb that provides essential illumination for the camera, and an exhaust fan that operates at the end of the disinfection cycle. A small air extraction system (Acerudobla 1″—6 W) was included to remove annoying reported UV-C lamp odors, such as thiols [37].

Since the development board operates at 3.3 V logic levels, level converters are used to interface with relays that switch 120 V AC power to the lamps and fan. To monitor the intensity of UV-C radiation, a pair of AS7331 sensors (SparkFun Electronics, Niwot, CO, USA) are used, one to monitor radiation from the superior lamp and the other to monitor radiation from the inferior lamp, integrated into a development board that communicates via I2C, ensuring low power consumption. In addition, an SDC30 sensor (Sensirion AG, Stäfa, Switzerland) is used to measure changes in temperature and humidity during the disinfection process; this sensor is also connected to the I2C port of the Arduino.

#### 2.4.3. The GUI on the Raspberry Pi

The UV-C seed disinfection system features a GUI powered by a Raspberry Pi 4 connected to a 7 inch touchscreen. A custom Python-based application provides control and monitoring capabilities, allowing users to configure the UV-C exposure time in minutes and view readings of temperature, humidity, and UV intensity. The system supports manual interruption and automatically stops the process when the timer is ended. Data are transmitted from the Arduino Nano 33 BLE (Arduino, Turin, Italy) to the Raspberry Pi via a serial connection. The application also supports the automatic generation of reports in Excel format, including sensor data recorded, date, time, and output graphs. Figure 5 displays the user interface window and the instructions for using the disinfection chamber.

#### 2.4.4. Computer Vision System for Color Monitoring

A compact Logitech C920 PRO HD (Logitech, Lausanne, Switzerland) webcam was installed in the chamber structure, with 1080p resolution and USB connectivity, to capture seed images during the disinfection process. A vision system was developed to capture and analyze webcam video frames in real-time, visualizing seed contours, color histograms, and chromaticity. Processing includes resizing, contour detection, and masking to highlight key regions. This system was implemented in Python (version 3.10.11) and processes each frame dynamically, displaying shape and color information through multiple interactive plots.

The GUI displays the webcam feed alongside three interactive plots as shown in Figure 6. A histogram in the LAB color space offers a clear view of the color distribution, revealing dominant tones, contrast, and uniformity in an image. In addition, visualizing LAB values on a chromaticity diagram further enhances analysis by showing how colors cluster or shift.

The pseudocode in Algorithm 1 illustrates the main functions of the implementation.
**Algorithm 1** Computer vision analysis application.1:**Start Application**2:Create GUI window3:Initialize video processing thread4:Display GUI5:**Inside VideoThread:**6:**while** application is running **do**7: Capture a frame from webcam8: Preprocess the image (resize, normalize)9: Detect contours and apply mask10: Generate LAB color histogram11: Update CIE chromaticity diagram12: Detect ArUco markers and estimate distance13: Combine all visualization images for GUI display14: Send final image to GUI15:**end while**16:**Inside GUI:**17:Show image on screen18:Handle user interactions:     • Save image     • Toggle average LAB calculation     • Close application safely

The system integrates OpenCV for real-time image capture and processing, Matplotlib (Version 3.9.3) and Seaborn (Version 0.13.2) for dynamic visualization, and Colour Science for precise chromaticity analysis in CIELAB and CIE 1931 spaces. The pipeline includes steps such as Canny edge detection to simplify images and accurately extract seed contours, ensuring color analysis focuses on relevant regions.

The CIELAB (LAB) color space represents all perceivable colors using three components: $L^{*}$ for lightness, $a^{*}$ for the red–green axis, and $b^{*}$ for the yellow–blue axis. Unlike device-dependent models, LAB is consistent across applications and perceptually uniform, making it highly accurate for color representation [38]. For comparison purposes and to assess the magnitude of color changes, the ΔE metric is employed. This metric quantifies the perceptual difference between a reference and detected color by computing the Euclidean distance between their coordinates in the CIELAB space, calculated as(1)$$\Delta E=\sqrt{{\left(L^{*}-L_{r}^{*}\right)}^{2}+{\left(a^{*}-a_{r}^{*}\right)}^{2}+{\left(b^{*}-b_{r}^{*}\right)}^{2}}$$
where $L^{*},a^{*},b^{*}$ represent the color values detected by the vision system and $L_{r}^{*},a_{r}^{*},b_{r}^{*}$ are the reference values. The ΔE metric is particularly suitable for evaluating seed color changes because it provides a perceptually uniform measure that aligns with human visual sensitivity, enabling the precise, quantitative assessment of subtle variations in seed coat color induced by UV-C treatment.

#### 2.4.5. Chamber Enclosure and Safety Design Considerations

The irradiation chamber is adapted using a commercial steel electrical box (VEVOR 40×30×15 cm), which serves as the primary structural enclosure to ensure durability and safety during operation. Inside the box, the custom-designed support structure securely holds the UV-C lamps in precise alignment, optimizing radiation coverage for effective seed disinfection. In addition, the electronic control system, including the microcontroller, sensors, camera, and power components, is neatly integrated within the chamber to maintain a compact and organized setup. As shown in Figure 7, the box has a tightly sealed door designed to provide an airtight environment, effectively containing UV-C radiation and preventing the escape of potentially annoying odors generated during the disinfection process, thus ensuring operator safety and environmental containment.

### 2.5. Experimental Evaluation and Iterative Refinement

The developed prototype is currently undergoing a rigorous testing and validation phase, with ongoing trials evaluating radiation uniformity, sensor accuracy, airflow effects, and ease of seed placement within the chamber. Iterative refinements are continuously implemented based on experimental seed trials and user feedback, ensuring improvements in safety, usability, and overall system performance. In parallel, controlled experiments with maize seed samples are being conducted to determine the agronomic benefits of UV-C priming. These trials assess key indicators such as germination rate, seed vigor, and viability, while also examining trade-offs in cost, operational efficiency, and adaptability to different production contexts. Such experimentation is essential for establishing whether UV-C exposure can reliably enhance seed performance and whether the system can be scaled for broader adoption in household- and community-level agriculture. The computer vision module was also tested as a proof of concept for tracking seed color changes during irradiation. The results of these evaluations are presented in the following section.

## 3. Results

### 3.1. Monitoring UV-C Irradiation, Temperature, and Humidity During Startup

Data acquisition was configured with a sampling rate of 2 s to monitor the behavior of the variables inside the chamber over a 10 min period. This timeframe was selected because UV-C lamps typically require a short stabilization period before reaching their maximum radiation intensity. The AS7331 and SDC30 sensors were placed in the center of the irradiation chamber, placed at similar distances from the lamps: 3 cm from the top lamp and 3 cm from the bottom lamp. Figure 8a presents the recorded data, illustrating the intensity stabilization curves for both lamps. The stabilization times of 1.1 min for the Top Lamp and 1.0 min for the Bottom Lamp indicate the points at which their radiation output consistently stays within a 6% tolerance range of their steady-state values.

Figure 8b shows the temperature and humidity levels inside the UV-C irradiation chamber during the 10 min test period. The temperature fluctuated between 23.4 °C and 28.6 °C, with an average of 26.3 °C, indicating a moderate but noticeable variation of about 5.1 °C. Humidity ranged from 46.91% to 60.17%, averaging 52.58%, with a wider variation of nearly 13.26%. These environmental changes suggest that, while the chamber conditions remained generally stable, some variation occurred that could potentially influence the performance of the UV-C lamp or the seed exposure.

### 3.2. Monitoring Variables Under Active Air Extraction Conditions

The air extractor alters the temperature and humidity within the chamber, potentially affecting seed conditions. These conditions are compared to the baseline values to highlight differences. As shown in Figure 9a, radiation is attenuated by 19.59% at the top and 13.25% at the bottom radiator.

Since all tests started under the same conditions, the air extraction system slowed the changes in temperature and humidity. Figure 9b compares both cases, highlighting the effect of the extractor on these variables. The temperature variation decreased by 1.3 °C and the humidity decreased by 4.6 percentage points compared to the baseline. This shows that the extractor not only improves air quality but also helps stabilize the microclimate of the chamber. Finally, Figure 9c presents the accumulated UV-C dose detected by the sensors over a 10 min period, clearly illustrating the impact of active air extraction. The results reveal a notable attenuation in accumulated radiation when the air extraction system is active, suggesting that airflow also plays a significant role in reducing UV-C exposure.

### 3.3. Monitoring Irradiation Through Cellophane

To position the seeds within the chamber during the UV-C irradiation process, a transparent cellophane container was used. This material was chosen because of its high transmittance, allowing UV-C radiation to pass through with minimal obstruction. However, since any material introduces some degree of attenuation, it was necessary to quantify the impact of cellophane on the UV-C intensity, particularly from the bottom lamp, since the seeds are placed directly above it. Measurements were carried out under these conditions to assess radiation loss as shown in Figure 10a. The results indicated a final attenuation of 17.20%, suggesting that while most UV-C radiation passes through, a modest reduction in intensity occurs. This attenuation must be taken into account when calculating the effective dose delivered to the seeds as illustrated by the accumulated dose graph in Figure 10b.

### 3.4. Uv-C Chamber Irradiation Mapping

To characterize the UV-C lamps, an irradiation mapping procedure was performed using a pair of AS7331 sensors. Measurements were taken for one minute at 15 positions along the X-axis of the irradiation tray, spaced 1.5 cm apart. The tray was then moved incrementally along the Y-axis, from 1 to 9 cm. At each point on the grid, the sensors recorded local irradiation levels, systematically covering the entire area. After data collection, the average values were calculated and combined into a single graphic. This graphic, shown in Figure 11, presents the resulting irradiation map, which illustrates the spatial distribution of UV-C radiation from both lamps, highlighting areas of peak intensity and overall coverage patterns.

The irradiation map provides a detailed spatial distribution of the UV-C intensity across the chamber, allowing the precise identification of areas with higher or lower radiation levels.

### 3.5. Evaluation of Computer Vision Performance in Seed Contour and Color Detection

To evaluate contour detection, tests were conducted on maize seeds of four different colors. Reliable detection was achieved only when sufficient contrast existed between the seeds and the background. Low contrast, uneven illumination, shadows, or image blurring often caused incomplete or failed detection. As shown in Figure 12, seeds with strong contrast were accurately outlined, while low-contrast samples produced suboptimal results.

Furthermore, to evaluate the precision of a computer vision program in detecting color within the CIELAB color space, six Post-it™ notes with bright colors were first measured using a handheld colorimeter (FRU WR-10QC) in a controlled laboratory setting to obtain the reference values. The same color samples were then analyzed using the computer vision system, which extracted the corresponding color values for each object as shown in Figure 13.

The ΔE metric was used to compare the colorimeter and vision system outputs, quantifying the magnitude of color differences. As shown in Table 3, notable discrepancies appear between reference and vision-based measurements, particularly for specific color tones. These differences can be attributed to factors such as uneven illumination, sensor calibration limits, and the lower spectral sensitivity of the camera compared to the precision of a dedicated colorimeter.

### 3.6. Experimentation with Maize Seed Samples to Determine the Benefits of the UV-C Radiation Process

White heirloom maize seeds were used for this study. They were harvested in November 2024 in the community of San Pedro Potla, Temascalcingo, State of Mexico, at an altitude of 2400 m above sea level. To ensure consistency under experimental conditions, the maize seeds were classified by size using two geometric square sieves measuring 8 mm and 7 mm. Seeds retained on the 8 mm sieve were categorized as large, while those retained on the 7 mm sieve were considered medium-sized. Medium-sized seeds were selected for use in this research. The type of maize, along with its physical dimensions and color, is detailed in Table 4.

Before sowing, selected maize seeds were treated using the UV-C radiation system. The seeds were placed inside the irradiation chamber shown in Figure 7 and exposed to radiation under two conditions: (1) with the air extractor turned off and (2) with the extractor turned on. A timer controlled the duration of the exposure, which was set at 10 min. The UV-C intensity, measured using a UV-C/254 measuring device, was 700 W/cm^2^.

#### 3.6.1. Fungal Infection Assessment (Infested Seed Count)

Fungal infection assessment was crucial to verify whether UV-C treatment reduces seed-borne pathogens, ensuring the process enhances both germination performance and sanitary seed quality. Thus, two independent germination processes were carried out according to the ISTA (2010) recommendations, based on the treatment conditions: (1) condition without extractor M5-T0 (control samples) and (2) M5-T1 (samples treated with UV-C for 10 min). Seeds (10 per box) were placed in sterilized plastic Petri dishes measuring 90 × 15 mm, using a layer of filter paper moistened with 6 mL of distilled water as substrate. Each treatment was replicated eight times. The boxes were kept in darkness, covered with aluminum foil, with an average temperature and humidity during the day of 24.8 °C and 31.71%, respectively. The number of infested seeds was determined at 84 and 96 h after the test was established. A seed was considered infested if it had at least one fungal colony. The experiments were conducted at the facilities of the Instituto Politecnico Nacional (IPN), at ESIME Zacatenco, Mexico City (19°29′56″ N, 99°08′06″ W) during the months of April and May, and observations were recorded 8 days after sowing.

Statistical analysis indicated significant differences (p<0.05) between the control samples and those treated with UV-C (10 min) in both experiments (samples treated without and with the air extractor). In both cases, a reduction was observed in the number of seeds infested with fungi as presented in Figure 14. The average percentage of infested seeds was 85% and 80% (untreated seeds), which decreased after treatment to an average of 65% and 59%, representing a reduction of 23.5% and 26.25%, respectively, showing a similar decreasing trend under both seed treatment conditions.

#### 3.6.2. Thermal Effects of UV-C Radiation on Maize Seeds

The radiation system not only emits UV-C light for disinfection but also generates heat, which affects the internal temperature of the chamber and the maize seeds themselves. As demonstrated by infrared camera images (FLIR IR i3), maize seeds experience a noticeable temperature increase during exposure. This heating effect is particularly important to monitor, as elevated temperatures can influence seed viability and moisture content, potentially affecting the germination results. Thermal images provide visual confirmation of the localized heat build-up on the seed surface, underscoring the need to consider thermal effects when designing and optimizing UV-C treatment protocols.

The sequence of images in Figure 15a–d illustrates the thermal behavior of maize seeds inside the UV-C irradiation chamber as captured with an infrared thermal camera. Before treatment, the seeds exhibited a uniform temperature distribution consistent with ambient conditions. Following UV-C exposure without air extraction, a clear rise in surface temperature was observed, confirming heat accumulation caused by radiation. In contrast, when the air extraction system was active, the seeds displayed ambient temperature levels prior to treatment and a noticeably smaller increase in temperature after exposure. This comparison demonstrates the extractor’s effectiveness in minimizing thermal accumulation and stabilizing the chamber’s microclimate during UV-C irradiation.

#### 3.6.3. Evaluation of Perceptible Color Differences in Maize Seeds After UV-C Radiation

UV-C exposure can cause some maize seeds to change color due to pigment degradation and cellular damage. The radiation breaks down DNA, proteins, and natural pigments like carotenoids and anthocyanins, leading to oxidative stress and visible changes such as browning, fading, or altered pigmentation. To assess color changes in maize seeds after UV-C exposure, a handheld colorimeter was used to measure two seed samples, M2 and M5, before and after irradiation. The average color difference (ΔE) was 2.4637 for M2 and 0.3469 for M5, indicating a more noticeable change in M2. Statistical analysis showed significant differences in all three CIELAB components for M2 (*p* < 0.002), reflecting changes in lightness and both color axes. In contrast, M5 exhibited a significant change only in the $a^{*}$ axis (*p* = 0.0008), with minimal variation in $L^{*}$ and $b^{*}$. These differences are shown in Figure 16a–c. According to the CIE76 standard [39], a ΔE above 2.3 is perceptible to the human eye, confirming that the color change in M2 is visually noticeable as illustrated in Figure 16d.

## 4. Discussion

This study presents a comprehensive design and evaluation of an automated UV-C irradiation system developed for the treatment of maize seeds, focusing on system performance, environmental stability, uniformity of irradiation, and seed response. The findings underscore the importance of both physical and environmental parameters in ensuring effective and consistent UV-C exposure for biological materials.

A key feature of the prototype is its use of accessible materials, such as coroplast and cellophane, chosen for their low cost and wide availability. Although more durable materials, such as stainless steel for the structure and quartz for seed containers, would offer greater longevity and structural integrity, the prototype was intentionally designed to support rapid deployment and community integration. These materials make the system more feasible for small producers and local adaptation. Although a stainless steel enclosure was considered, the design remains flexible and can be adapted to any locally available structure. This adaptability empowers end users, including researchers and small-scale farmers, to tailor the system to their specific needs and local context, fostering sustainable and community-driven implementation.

### 4.1. Advancing Prototype Design Through Sensor-Based Feedback

The results from the system characterization underscore the critical need for environmental monitoring inside the UV-C chamber. Unlike the previous prototypes evaluated by Méndez et al. [31], which lacked any instrumentation, the current system integrates sensors to provide real-time data on UV-C intensity, temperature, and humidity. In the previous designs, there was no knowledge of radiation distribution or environmental changes within the chamber, making it impossible to evaluate how these factors might influence treatment efficacy.

The system characterization phase showed that both UV-C lamps achieved stable radiation output in approximately one minute, consistent with the warm-up behavior reported in similar systems [40]. Previous prototypes did not account for the warm-up behavior of the lamps, which can affect the actual exposure time received by the seeds. While thermal and humidity variations were moderate, they underscore the impact of the closed-chamber environment on internal microclimate conditions. The inclusion of an air extraction system effectively reduced undesirable lamp-generated odors and stabilized temperature and humidity. However, this also caused a slight reduction in UV-C intensity, which could impact treatment efficacy. Further research is needed to evaluate how stabilized environmental conditions affect seed response, particularly regarding thermal stress and moisture retention. Balancing environmental control with effective radiation exposure remains a critical factor in optimizing system performance.

The spatial mapping of UV-C intensity across the chamber floor revealed a non-uniform distribution, with identifiable high- and low-radiation zones. The current mapping, however, was limited to a single vertical axis at the chamber center. Full characterization requires additional measurements across lateral and diagonal positions to generate a complete spatial profile, which is essential for optimizing seed placement and ensuring uniform exposure. The evaluation of radiation transmission through the cellophane seed holder highlighted the importance of accounting for material attenuation in dose calculations. Although the measured 17.2% reduction in intensity falls within acceptable limits for transparent materials, neglecting it could lead to underdosing if not properly incorporated into system calibration.

The irradiation map obtained in this study provides essential information for optimizing seed placement, ensuring uniform and sufficient UV-C exposure—a factor neglected in earlier prototypes, where seeds were positioned without considering spatial dose variation. Previous designs, such as the prototype presented by Hernandez-Aguilar et al. [20], relied on symmetric lamp placement and reflective walls to assume complete radiation coverage, without verifying actual distribution. This could result in suboptimal doses for certain seeds, causing inconsistent treatment. In contrast, the current system combines top and bottom lamps with integrated sensors to measure real-time radiation at multiple positions, enabling the precise mapping of UV-C intensity and consistent dosing for every seed. Additionally, UV-C sensors allow accurate assessment of radiation attenuation through materials like cellophane, supporting precise dose calculations. This comprehensive monitoring ensures reproducibility, safety, and optimized treatment outcomes, directly addressing critical gaps in earlier prototypes.

### 4.2. Computer Vision Proof-of-Concept and Future Integration

Computer vision analysis demonstrated the system’s ability to detect seed contours and monitor color changes, although performance varied with contrast and lighting conditions. Contours were reliably detected when seeds had strong contrast against the background, but low contrast, uneven illumination, shadows, or image blurring often resulted in incomplete or failed detection. Algorithms such as Canny rely on pronounced intensity gradients, and thresholding may cause seeds to blend with the background when pixel values are similar. Gaussian blurring is used to reduce noise and improve edge detection, but excessive blurring or poor lighting still limits accuracy. Discrepancies between reference and detected colors, quantified via ΔE in CIELAB space, indicate that calibration or more robust segmentation and color correction may be needed.

Improvements include optimizing chamber lighting to enhance contrast and reduce shadows, applying advanced preprocessing like adaptive histogram equalization, and using robust segmentation algorithms, such as machine learning models. Periodic color calibration and correction can also reduce ΔE discrepancies, enhancing contour detection and color monitoring accuracy. However, the current computer vision implementation was selected as a proof-of-concept because it operates without requiring retraining for each prototype modification, allowing rapid adaptation to design changes. This flexible approach lays the groundwork for future integration of machine learning, which could enhance segmentation, handle low-contrast or unevenly lit samples, and enable the more precise, automated analysis of seed color and quality.

### 4.3. Effectiveness of UV-C Treatment on Fungal Reduction in Maize Seeds

Experimental treatment of maize seeds with UV-C radiation for 10 min, under both active and inactive air extraction conditions, resulted in significant reductions in fungal infestation. A 23.5–26.25% decrease in contaminated seeds highlights the potential of UV-C as a non-chemical disinfection method and a practical alternative to fungicides, particularly for small-scale or organic production. Thermal imaging confirmed the appreciable heating of seed surfaces during exposure, underscoring the need to manage thermal effects to avoid damage. While these results demonstrate the prototype’s effectiveness in reducing fungal contamination, further optimization is required to improve radiation uniformity, environmental control, and visual monitoring. Ongoing research is also necessary to assess long-term impacts on seed germination, vigor, and nutritional quality, and to refine treatment protocols based on UV-C intensity and duration.

### 4.4. Future Directions in High-Capacity UV-C Seed Sterilization

When working with solid samples, irradiation conditions that promote a homogeneous distribution of UV-C across the entire surface are essential to maximize treatment efficacy [41]. Effectiveness depends not only on radiation parameters but also on the characteristics of both the sample and the container. For example, the ability of the container to allow air extraction can influence whether UV-C is evenly distributed and whether heating occurs, potentially creating synergistic effects that enhance treatment outcomes [42,43].

In the present system, several design considerations were implemented: a container that holds a single seed layer, and an irradiation unit with UV-C sources located above and below, enabling bidirectional exposure. Maize seeds were placed in a transparent cellophane holder, which minimally absorbs UV-C, allowing both sides of the seed to be irradiated. This configuration enabled the results reported in the current study.

Other studies have sought to improve irradiation uniformity through multiple light sources, sample movement, or conveyor systems for short exposure times [17]. However, conveyor-based approaches often result in limited UV-C penetration, restricting sterilization primarily to the seed surface when seeds remain static, i.e., without rotation. For instance, Hidaka and Kubota [44] exposed wheat seeds to UV-C on a conveyor system with upper and lower lamps; seeds were irradiated for 4 s per cycle at 97 W/m^2^ and 2 cm from the light source, but sterilization was limited to exposed surfaces. Similarly, Kamel et al. [17] used a hopper system feeding seeds onto a conveyor belt, irradiating different samples at 10.5 mW/cm^2^ for 5–45 min.

By contrast, the present system achieved bidirectional exposure because UV-C permeability through the fixed transparent cellophane band allowed irradiation on both the upper and lower seed surfaces, although with measurable attenuation. The dual-lamp fixed configuration allows short to extended exposures (seconds to hours) at consistent intensity, making it well-suited for microgreen production, with a capacity aligned to the 20–25 g per variety typically used weekly in greenhouse cultivation.

For field-scale maize production, however, larger community-level irradiation systems would be needed, as farmers in San Pedro Potla (State of Mexico) report requirements of 20–25 kg of maize seed per hectare, corresponding to a planting density of 50,000–55,000 plants/ha [45]. Future designs should therefore consider higher-capacity UV-C chambers. Mechanical rotary systems may be preferable for bulk seed disinfection, as conveyor belts may limit efficiency. Studies with peppercorns, cumin, and clove seeds have shown that seeds do not create “shadowing” or clouding effects that hinder UV-C exposure. Moreover, the robust physical structure of seeds allows them to slide and rotate without damage, facilitating uniform irradiation [46,47,48].

### 4.5. Feedback Control and Regulatory Strategies for UV-C Disinfection

Effective UV-C seed disinfection requires precise radiation delivery coupled with continuous monitoring to ensure reproducibility, safety, and treatment efficacy. While the current prototype lamps do not allow the dynamic adjustment of intensity, the UV-C dose can be controlled by regulating the distance between the seeds and the lamps. Sensor-based feedback control systems allow dynamic adjustment of exposure duration, intensity, and airflow based on real-time measurements of UV-C dose, temperature, and humidity. Such adaptive control prevents under- or over-exposure, mitigates thermal stress, and enables optimized protocols for different seed types and treatment objectives.

From a regulatory perspective, international frameworks provide guidance for the safe and effective use of UV-C technologies. For example, the U.S. Environmental Protection Agency (EPA) [49] establishes guidelines for UV water treatment systems, defining acceptable dose ranges, monitoring requirements, and safety measures. Similarly, the International Ultraviolet Association (IUVA) [50] publishes standards and recommendations for UV disinfection applications, including safe lamp operation, exposure limits, and performance verification procedures. ISO standards, such as ISO 15858:2016 [51], specify safety requirements for UV-C devices used in germicidal applications, addressing both human exposure limits and engineering controls.

Although these regulations primarily address water and surface disinfection, their principles can inform seed treatment protocols. Key considerations include ensuring uniform dose delivery, monitoring environmental conditions, implementing radiation containment, and protecting users through engineering and procedural controls. Integrating these international norms into prototype design enhances safety, reproducibility, and the potential for regulatory compliance, supporting the wider adoption of UV-C seed disinfection in research and commercial agriculture.

### 4.6. UV-C Radiation and Its Influence on Seed Surface Appearance and Pigment Degradation

In this study, changes were observed in the color components of the maize seeds. Other authors have similarly reported that UV-C radiation induces alterations in color-related variables in agricultural seeds and food products. Hernandez et al. [21] found changes in an indirect measure of color through photoacoustic absorption spectra (wavelength range 300–700 nm) in common bean seeds (*Phaseolus vulgaris* L.) irradiated with UV-C for 2, 5, 10, and 15 min. These changes were inferred from the decrease in signal at various wavelengths and corroborated using micrographs that compared irradiated and non-irradiated seeds. Their findings showed a greater degradation of the seed surface with increased exposure to UV-C.

In pinto beans (*Phaseolus vulgaris* L.), Junk-Knievel et al. [52] observed a gradual decrease in brightness ($L^{*}$) over time as exposure to UV-C radiation increased. Similarly, Kumar et al. [53] reported color changes in pea seeds (*Pisum sativum* L.) exposed to UV-C radiation for 5, 10, and 20 min and subsequently stored at temperatures of 15, 20, and 25 °C. Over time, seed color changes were evident, with the condition showing the least color degradation and the greatest stability, particularly in terms of brightness ($L^{*}$) and chroma values, with 20 min of exposure to UV-C followed by storage at 15 °C. This indicates that post-irradiation storage conditions may play a role in preserving seed quality.

Color changes have also been evaluated using the ΔE metric, where values below 2 represent a small difference that is imperceptible to the human eye [54]. In the present study, the values of ΔE ranged from 7 to 13 in the lower range and 20 to 26 in the higher range, suggesting that these differences were perceptible as changes in surface opacity. The degree of color change varied by seed type. It should be noted that other authors, such as [55,56], reported no perceptible color changes in black pepper (*Piper nigrum* L.) when applying UV-C treatment for pasteurization and decontamination purposes.

The use of a camera for the real-time monitoring of maize seed color under UV-C irradiation is innovative because it enables the direct, non-destructive tracking of pigment degradation and surface appearance changes. While previous studies have reported color changes in seeds such as beans and peas using indirect methods like photoacoustic spectra or post-irradiation micrographs [21,52,53], our approach allows the continuous, automated quantification of color variation on individual seeds using metrics such as ΔE. This real-time imaging provides more immediate and precise feedback on UV-C effects, improving the ability to optimize exposure parameters and assess treatment efficacy directly on maize seeds—a capability not previously demonstrated in the literature.

### 4.7. System Limitations and Pathways for Regulatory-Compliant Improvement

The system presents several limitations that must be addressed to enhance its effectiveness, safety, and scalability. A primary design constraint for industrial or large-scale applications is the use of plastic materials, which lack the durability and resilience required for long-term or high-throughput operation. Future iterations should incorporate more robust materials and structural reinforcements to meet the demands of large-scale seed treatment.

Measurement accuracy is limited by the absence of sensor calibration and lack of comparison with standardized equipment. Image acquisition also requires improvement, including enhanced contrast, optimized thresholding, reduced blur, and uniform lighting. Additionally, the integration of presence sensors is essential for operational safety, preventing lamp activation when the lid is open or personnel are nearby, in accordance with international UV-C safety guidelines such as ISO 15858:2016 on UV-C devices for germicidal applications.

To improve flexibility and efficiency, particularly in industrial contexts, the system should enable remote monitoring and control via Bluetooth or Wi-Fi—capabilities already supported by the current hardware. Implementing these features would not only enhance operational performance but also facilitate compliance with international standards for UV-C radiation exposure, device safety, and user protection, aligning the prototype with best practices in both food and seed disinfection applications.

## 5. Conclusions

This study presented the design, characterization, and performance evaluation of an automated UV-C irradiation system for maize seed treatment, focusing on disinfection efficacy, environmental control, and integration of a computer vision monitoring system. The system achieved a 23.5–26.25% reduction in fungal infestation after a 10 min exposure, confirming the potential of UV-C radiation as an effective and non-chemical alternative to conventional fungicides, particularly for small-scale or organic farming applications.

The UV-C lamps used reached stable output in one minute, while moderate fluctuations in temperature and humidity highlighted the influence of the closed-chamber environment. The addition of an air extraction system improved thermal stability and mitigated heat accumulation but also reduced UV-C intensity, a trade-off that must be considered during treatment planning. Radiation mapping revealed spatial non-uniformity in intensity, with current data limited to a central vertical axis; future measurements across multiple planes are needed for full spatial characterization. A 17.2% attenuation through cellophane seed holders further emphasized the need to account for material properties in dose calculations. The computer vision system demonstrated utility in detecting seed contours and color changes but was sensitive to lighting and contrast, indicating the need for improved calibration and image processing.

Limitations such as the reliance on plastic materials, the lack of sensor calibration, the absence of safety mechanisms, and the need for remote operation were identified, particularly for the scaling to industrial use. In general, the study establishes a foundational framework for UV-C seed treatment and highlights key areas for refinement, including dose uniformity, environmental regulation, and monitoring accuracy. Future research should assess the long-term effects of UV-C exposure on germination, plant development, and seed physiology to support broader agricultural adoption.

## Figures and Tables

**Figure 1 sensors-25-06070-f001:**
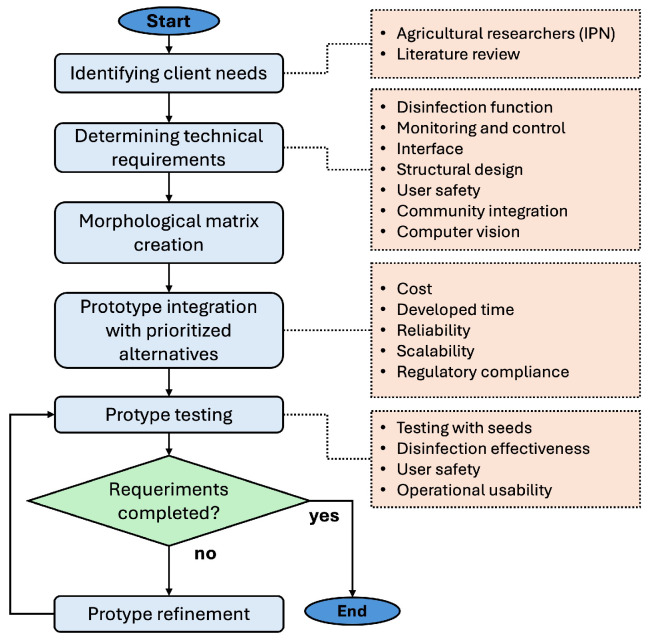
Design workflow of the automated UV-C irradiation system, illustrating the sequential stages from conceptualization and prototyping to integration and continuous performance evaluation for refinement.

**Figure 2 sensors-25-06070-f002:**
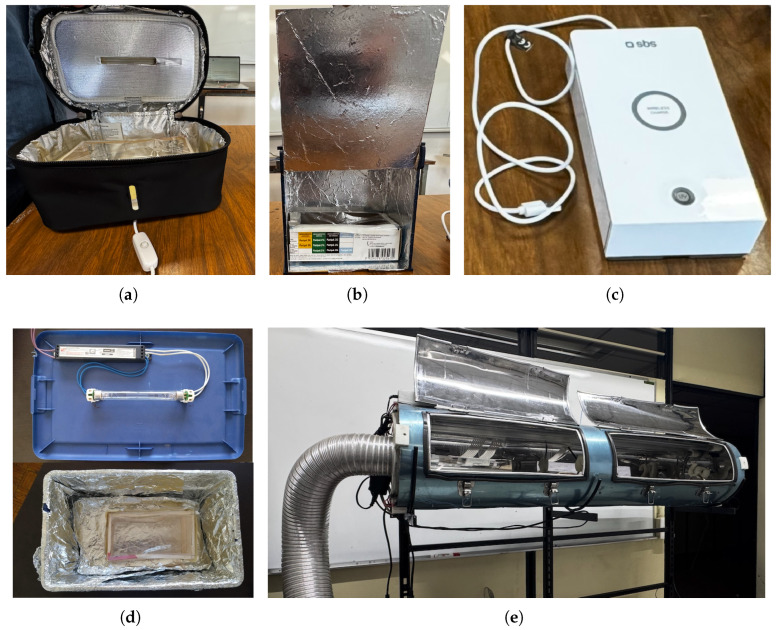
UV-C Irradiator Systems Implemented at ESIME-Zacatenco, IPN: (**a**) Bright Solution STERI BAG, a compact portable pouch for pilot seed tests; (**b**) wooden desktop box made of MDF and fitted with UV-C LEDs for small-scale use; (**c**) SBS commercial sterilizer combining UV-C disinfection with wireless charging; (**d**) plastic tote retrofitted with a Philips TUV 36 W lamp for medium seed batches; (**e**) large stainless-steel cylinder with multiple UV-C lamps and a rotating drum for industrial-scale treatments.

**Figure 3 sensors-25-06070-f003:**
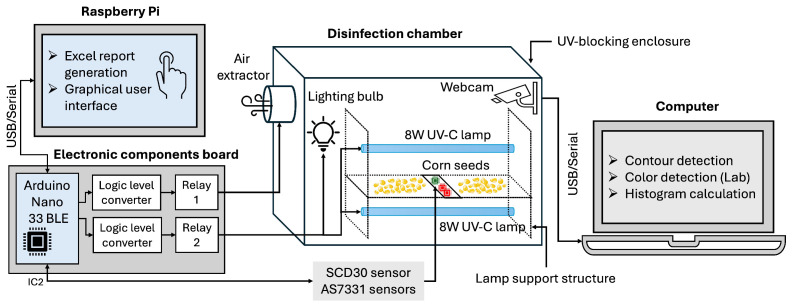
Comprehensive architecture of the automated UV-C irradiation system, showing the disinfection chamber, electronic control units, sensors, actuators, and air extraction integration.

**Figure 4 sensors-25-06070-f004:**
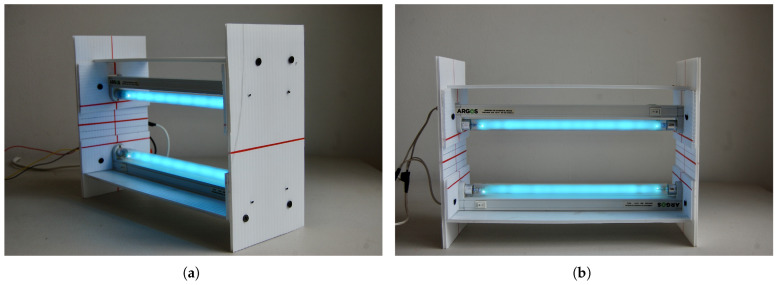
Adjustable support structure for ultraviolet-C lamps, showing (**a**) the side view with modular panel assembly and (**b**) the corner view highlighting the portability and stability of the design.

**Figure 5 sensors-25-06070-f005:**
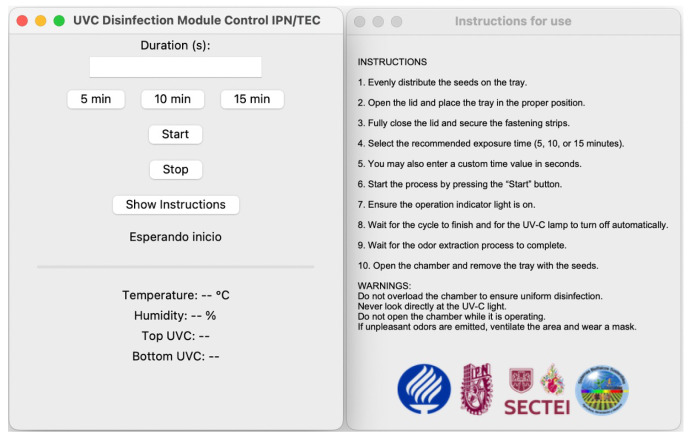
Graphical user interface developed in Raspberry Pi for the real-time control of irradiation timing, monitoring of environmental variables, and safety alerts.

**Figure 6 sensors-25-06070-f006:**
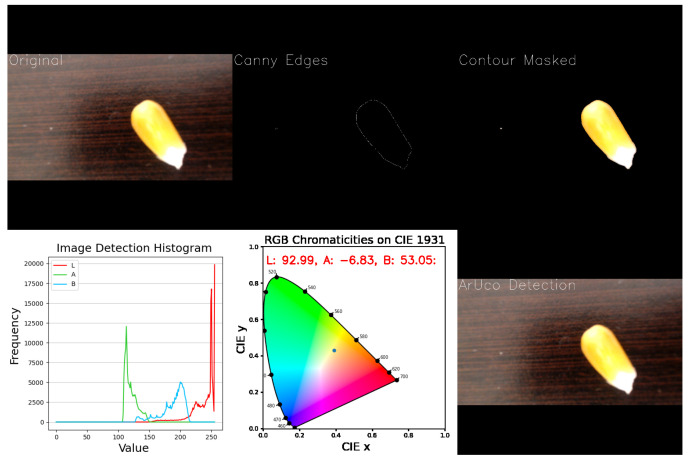
Desktop-based graphical user interface of the computer vision module, enabling the detection of seed contours and color changes for automated quality assessment.

**Figure 7 sensors-25-06070-f007:**
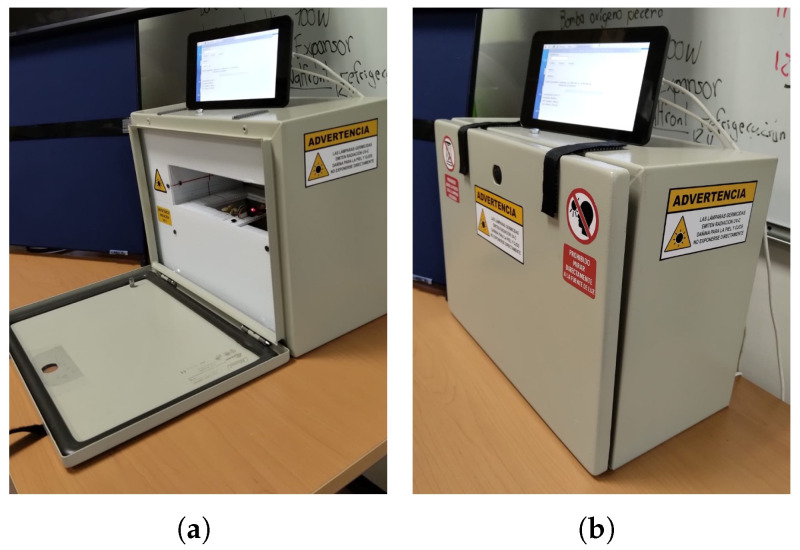
Enclosure of the irradiation system, employed to ensure user safety, prevent light leakage, and maintain controlled internal conditions during operation. (**a**) Image with the door open, showing the interior of the system; (**b**) Image with the door closed.

**Figure 8 sensors-25-06070-f008:**
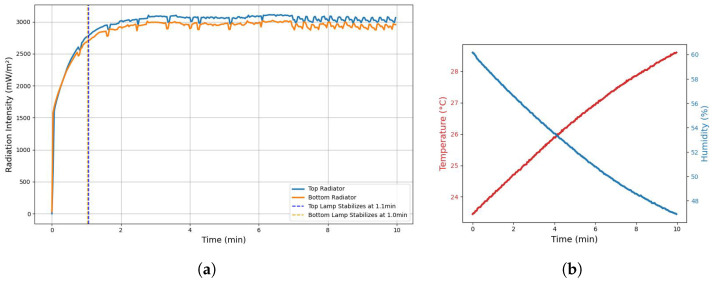
Startup characterization of the chamber over time, showing (**a**) the stabilization of UV-C lamp intensity and (**b**) the corresponding rise in internal temperature and decrease in humidity.

**Figure 9 sensors-25-06070-f009:**
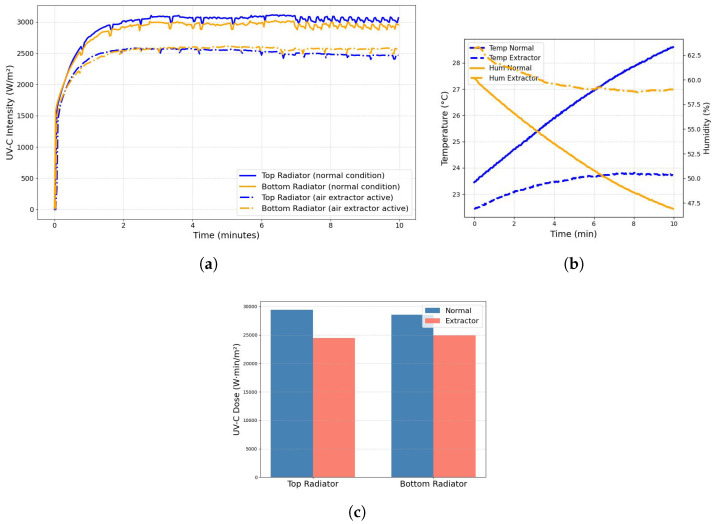
Comparison of startup conditions with and without air extraction: (**a**) UV-C lamps intensity stabilization, (**b**) temperature and humidity trends, and (**c**) differences in accumulated UV-C dose.

**Figure 10 sensors-25-06070-f010:**
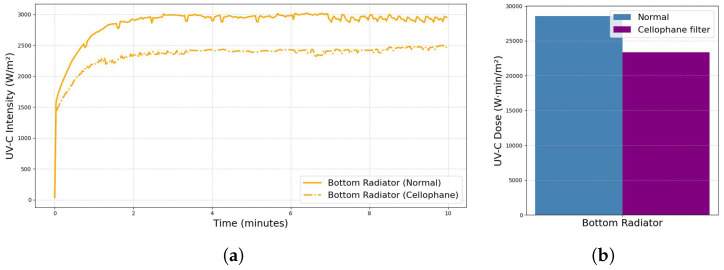
Comparison of startup conditions with and without cellophane film: (**a**) UV-C lamp intensity stabilization and (**b**) differences in accumulated UV-C dose due to attenuation by the barrier.

**Figure 11 sensors-25-06070-f011:**
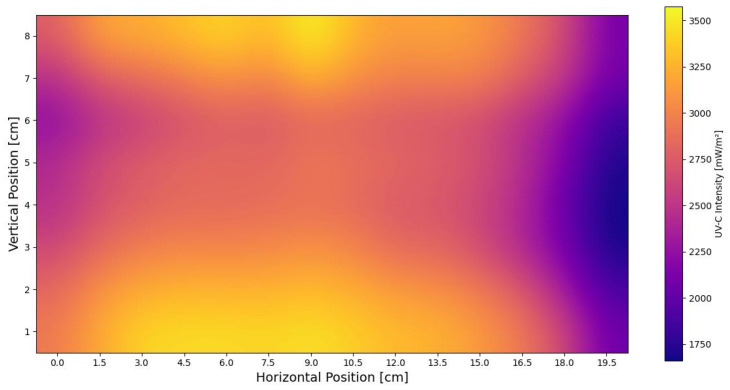
Combined ultraviolet-C radiation map integrating contributions from the superior and inferior lamps, revealing spatial distribution patterns and non-uniformity within the chamber.

**Figure 12 sensors-25-06070-f012:**
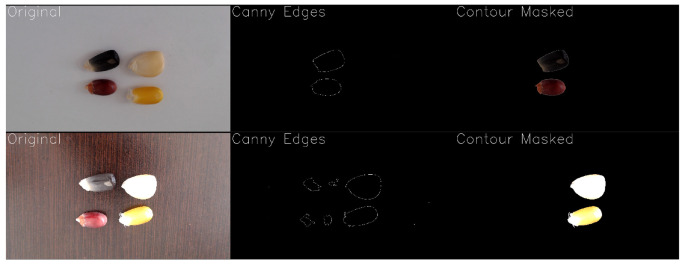
Detail of the vision GUI showing different conditions for the seeds samples and their changing results. The images illustrate how changes in scene setup affect edge detection performance.

**Figure 13 sensors-25-06070-f013:**
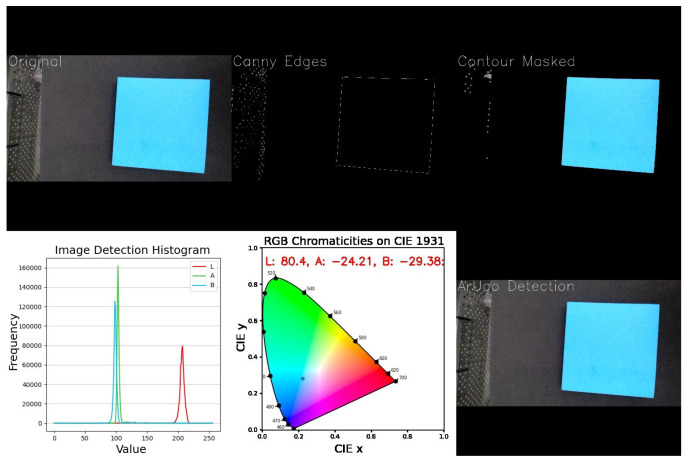
A bright-colored seed sample was detected by the vision system and used as a reference for comparison with colorimeter measurements.

**Figure 14 sensors-25-06070-f014:**
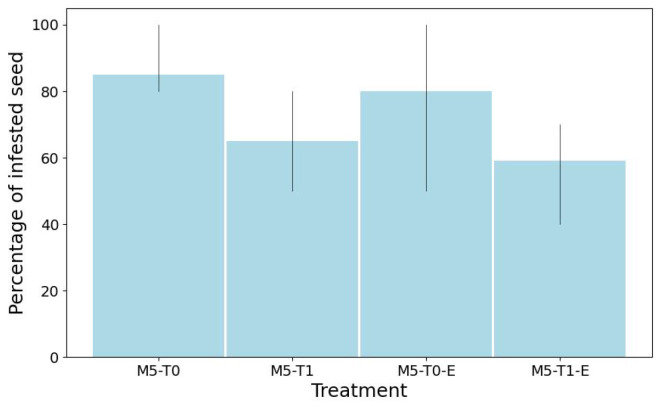
Percentage of infested seeds. M5-T0 and M5-T0-E: control samples from independent experiments; M5-T1: samples treated with UV-C radiation (10 min) without the air extractor turned on; M5-T1-E: samples treated with UV-C radiation (10 min) with the air extractor turned on (*p* < 0.05).

**Figure 15 sensors-25-06070-f015:**
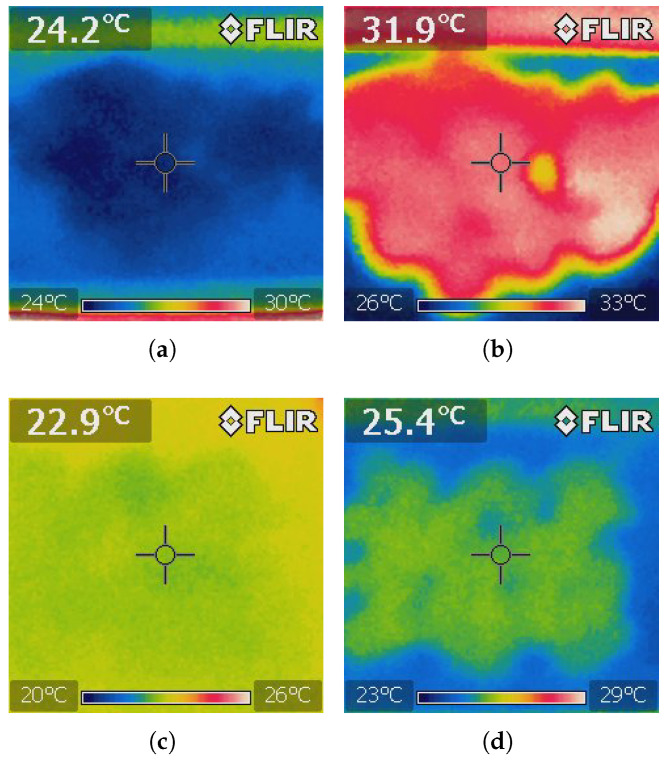
Infrared images of maize seeds before and after UV-C treatment: (**a**) Before treatment in normal condition. (**b**) After UV-C exposure in normal condition. (**c**) Before treatment with air extraction. (**d**) After UV-C exposure with air extraction.

**Figure 16 sensors-25-06070-f016:**
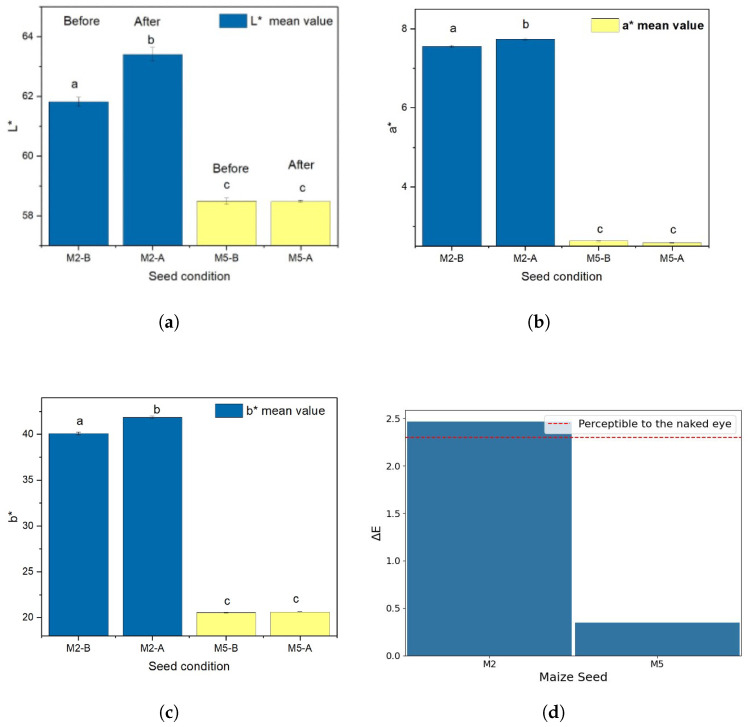
Comparing color shifts in maize seeds M2 and M5 under UV-C exposure: (**a**) Changes in lightness ($L^{*}$); (**b**) Changes in red–green coordinate ($a^{*}$); (**c**) Changes in yellow–blue coordinate ($b^{*}$). Statistical differences between treatments were determined using Tukey’s test, with significant groupings indicated in each subplot. (**d**) Average ΔE by seed type, representing overall color change.

**Table 1 sensors-25-06070-t001:** Morphological matrix presenting the design alternatives considered for the system, providing the basis for systematic selection.

	Feature/Solution	Alternative 1	Alternative 2	Alternative 3
1	Generate UV-C Radiation Lamps	Low-pressure Mercury	High-pressure Mercury	UV-C LED array
2	Control Radiation Intensity	PWM dimming	Variable power supply	Lamp distance adjustment
3	Monitor UV-C Intensity	Photodiode	Digital sensors	Light meter
4	Monitor Temperature	DHT22 sensor	Thermocouple	Digital sensor
5	Monitor Humidity	DHT22 sensor	Digital sensor	Capacitive Sensor
6	UV-blocking enclosure	Full-metal box	Opaque plastic housing	Custom composite material
7	Provide real-time data interface	Raspberry Pi and Touchscreen LCD	Web dashboard	Mobile app
8	Enable visual seed analysis	HD webcam	Raspberry Pi camera	Industrial camera
9	Adjust exposure time	Digital Timer	Manual Knob	Programmable microcontroller
10	UV-C Lamp support structure	Screwed panels	Snap-fit parts	Magnetic fixtures
11	Enable Chamber portability	Wheels	Handle and lightweight	Modular compact design
12	Handle multiple seed samples	Multiple trays	Rotating tray system	Slide-in drawer trays
13	Facilitate odor extraction	Mini exhaust fan	Charcoal air scrubber	Air filter installation
14	Adapt to a variety of UV-C enclosures	Modular Mounting frame	Telescopic Sliding Rack Units	Clip-on module panels
15	Optimize Energy Use	Auto-off function	Low-power electronics	High-efficiency PSU
16	Ensure affordability	Open-source hardware/software	Standardized off-the-shelf components	Simplified modular design
17	Community-friendly interface and operation	Large buttons and icons	Simple mobile interface	Multilingual support
18	Design for easy repair	Quick-swap modular parts	Color-coded internal parts	Snap-fit modular design
19	Capture images for seed documentation	Periodic image capture	Comparison camera	Integrated logger
20	Seed detection with computer vision	Blob detection	YOLO-based detection	Edge detection algorithms
21	Visual/sound indicator of successful treatment	Beep signal	Green indicator LED	End sound and display
22	User customization options	Adjustable cycle presets	Profile-based settings	Touchscreen wizard setup
23	Durable, UV- and corrosion-resistant materials	UV-resistant plastics	Anodized aluminium	Stainless steel + coatings
24	Power supply options	AC mains (110–220V)	Solar panel + inverter	Hybrid system
25	Automated report generation	Auto-save to connected storage	PDF report generation	Cloud-based reporting
26	Automatic seed color analysis	RGB camera with calibrated lighting	Multispectral imaging	Open-source vision module

**Table 2 sensors-25-06070-t002:** Prioritized design alternatives selected from the morphological matrix, showing the criteria used for decision-making and their relative importance to the system’s overall performance.

Feature	Selected Alternative ^1^	Cost	DT	Reliab.	Scalab.	RC
1	Low-pressure Mercury	+	+	0	0	+
2	Lamp Distance Adjustment	0	0	+	0	+
3	Digital UV-C sensors	0	0	+	0	+
4	Digital sensor	0	0	+	0	+
5	Digital Sensor	0	0	+	0	+
6	Full metal box	0	+	+	+	+
7	Raspberry Pi and Touchscreen LCD	0	0	+	0	+
8	HD Webcam	-	0	+	0	+
9	Programmable microcontroller	-	0	+	+	0
10	Screwed panels	-	-	0	0	0
11	Handle and lightweight	-	-	+	+	0
12	Multiple trays	-	-	+	+	0
13	Mini exhaust fan	-	-	0	+	0
14	Modular mounting frame	0	0	+	+	+
15	Low-power electronics	0	0	+	+	+
16	Open Source Hardware and Software	+	0	0	+	0
17	Modular Standard Components	+	0	+	+	0
18	Quick-swap modular parts	0	-	+	+	0
19	Periodic Image Capture	0	0	+	+	0
20	Edge Detection algorithms	+	-	0	+	0
21	End sound and display	+	+	0	+	0
22	Adjustable Cycle Presets	0	-	+	+	0
23	Stainless Steel and Coatings	-	0	+	0	+
24	AC mains (110–220 V)	+	+	+	+	0
25	Auto-save to connected storage	0	0	+	+	0
26	Open-source vision module	+	0	0	+	0

^1^ Here, F stands for Feature, C for Cost, DT for Reliability, S for Scalability, and RC for Regulatory Compliance. The symbols represent the impact on each criterion: (+) indicates a positive effect, (0) a neutral effect, and (-) a negative effect.

**Table 3 sensors-25-06070-t003:** Comparison of LAB data between colorimeter and vision system.

Object	Colorimeter			Computer Vision			ΔE ^1^
	$L^{*}$	$a^{*}$	$b^{*}$	$L^{*}$	$a^{*}$	$b^{*}$	
C1	71.09	22.73	—16.53	74.39	21.21	—4.01	13.01
C2	75.07	45.74	54.16	78.20	19.79	57.36	26.64
C3	78.08	—21.96	38.26	79.53	—17.96	31.95	7.47
C4	90.73	—2.86	59.85	98.13	—11.73	38.54	24.36
C5	65.80	53.84	2.02	71.09	43.52	16.83	20.10
C6	63.79	—25.08	—22.78	80.40	—24.21	—29.38	17.94

^1^ ΔE is the color difference between the colorimeter and vision system in CIELAB space.

**Table 4 sensors-25-06070-t004:** Characteristics of the maize seeds used.

Type	Color (CIELAB)			Dimensions (mm)		
	$L^{*}$	$a^{*}$	$b^{*}$	**Length**	**Width**	**Thickness**
Semi-crystalline	5.95±3.93	3.14±0.52	17.77±1.05	11.12±0.63	6.13±0.63	4.49±0.87

## Data Availability

No new data were created or analyzed in this study.

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
