# Peer review of "Design and Evaluation of an Automated Ultraviolet-C Irradiation System for Maize Seed Disinfection and Monitoring"

_sensors, 2025, doi:10.3390/s25196070_

Round 1
Reviewer 1 Report
Comments and Suggestions for Authors
The paper focuses on the issue of seed disinfection and designed a UV radiation disinfection device, which has functions such as disinfection light adjustment, temperature and humidity and radiation illumination perception, and seed visual detection. The designed system has good practicality.
There are still some shortcomings in the paper.
- In the designing procedure, there is a lack of requirements or constraints regarding the seed processing capacity, efficiency, and other aspects of the device. How large does the seed collection box need to be and what is the seed filling thickness? What structural improvements are needed for continuous seed disinfection?
- The image recognition method in the paper lacks clear instructions on its intended use. The image recognition examples in the paper should provide a comparison of the original images under different conditions, and provide appropriate explanations for the possibility of recognition.
- The feedback control of disinfection in the paper should be discussed in more depth, and possible regulatory schemes should be explained.
- There are some formatting deficiencies in the paper, such as L157 "Table??.”; L269“1.37°C C”.
Author Response
Comment 1: In the designing procedure, there is a lack of requirements or constraints regarding the seed processing capacity, efficiency, and other aspects of the device. How large does the seed collection box need to be and what is the seed filling thickness? What structural improvements are needed for continuous seed disinfection?
Response 1: Thank you for your valuable observation. It highlights an important aspect that was omitted in the original version and helps us to clarify a key point. In line 140, we have added a brief paragraph explaining that the chamber size was designed to align with the needs of microgreen production, as small-scale producers are the primary target users of this prototype. We now also specify the typical weekly seed quantities they handle and emphasize that they require disinfection in small cycles.
Furthermore, regarding structural improvements for continuous seed disinfection, this has been addressed in the discussion section (line 553). There, we explain that if large-scale production is required to treat greater seed volumes, alternative approaches such as mechanical rotary systems and conveyor belts could be implemented to enable continuous processing.
Comments 2: The image recognition method in the paper lacks clear instructions on its intended use. The image recognition examples in the paper should provide a comparison of the original images under different conditions, and provide appropriate explanations for the possibility of recognition.
Response 2: Thank you for this insightful observation. We recognize that the previous version of the manuscript provided a weak explanation of the vision system, which may have led to ambiguity regarding its role and current development stage. In the revised version, we have clarified that the vision system was implemented as an early-stage proof-of-concept to demonstrate the feasibility of integrating artificial vision into seed disinfection prototypes. At this stage, only preliminary tests have been conducted, but its inclusion highlights the potential of such approaches for future optimization.
To strengthen this point, we emphasized the relevance of the vision system in the introduction by citing key studies on computer vision applications in maize seeds (line 69) and explaining its importance within the context of our prototype (line 98). Additionally, the manuscript has been reorganized to improve clarity, ensuring that the methodology and results sections now provide a more structured and coherent explanation of the vision system and its current outcomes.
Finally, in the discussion section (line 493), we expanded our remarks on the role of computer vision, outlining both the improvements required and the planned future developments within the ongoing research, including more robust segmentation, calibration, and potentially machine learning-based enhancements.
Comments 3: The feedback control of disinfection in the paper should be discussed in more depth, and possible regulatory schemes should be explained.
Response 3: Thank you for your valuable comment. We agree that the first version did not sufficiently elaborate on feedback control and regulatory schemes for UV-C disinfection. In the revised manuscript, we expanded the Discussion Section in line 562 to provide a more in-depth explanation. Specifically, we clarified that although the current lamps cannot dynamically adjust intensity, the UV-C dose can be regulated by modifying the distance between the seeds and the lamps. We also discussed the role of sensor-based feedback systems for real-time control of dose, temperature, and humidity, highlighting their importance in preventing under- or over-exposure and ensuring reproducible treatments.
Additionally, we incorporated international regulatory frameworks to contextualize safety and compliance. References to U.S. EPA guidelines, IUVA recommendations, and ISO standards (e.g., ISO 15858:2016) were included to illustrate how established principles in water and surface disinfection can inform the development of protocols for seed treatment. These additions address your concern by providing both the technical perspective of adaptive feedback control and the broader regulatory context, underscoring how future iterations of the prototype can align with internationally recognized safety and performance standards.
Comments 4: There are some formatting deficiencies in the paper, such as L157 "Table??.”; L269“1.37°C C”.
Response 4: Thank you for your observations regarding the format. The deficiencies have been corrected in this version.
Reviewer 2 Report
Comments and Suggestions for Authors
This manuscript reported the design, characterization and performance evaluation of an automated UV-C irradiation system for maize seed treatment, focusing on disinfection efficacy, environmental control, and integration of a computer vision monitoring system. This paper has not a high novelty from the view of sensor development. However, it put affordable sensors and circuit board together to achieve a common goal. The paper is well written, and I only have a few comments showing below:
There is no Figure 1f.
L157. Which table?
L245 to 260. Can your temperature and relative sensors measure 2 digital numbers? Do they have 2 digital accuracies? If not, one digital number is enough.
Author Response
Comment 1: There is no Figure 1f. The correction was done, the correct name was figure 2
Response 1: Thank you for the observation. The correction has been made, and the figures have been properly arranged in this revised version.
Comment 2: L245 to 260. Can your temperature and relative sensors measure 2 digital numbers? Do they have 2 digital accuracies? If not, one digital number is enough. Values have been rounded
Response 2: Your comment was very pertinent, as we initially reported sensor accuracies with varying levels of precision. In this revised version, we have rounded the values to ensure consistency throughout the manuscript.
Reviewer 3 Report
Comments and Suggestions for Authors
-
The literature review has not adequately covered the novelty of this work.
-
In the introduction, explain what you applied and what was done on corn, including the use of a camera.
-
Explain how innovative it is to use a camera for monitoring the color of corn in this study.
-
In the introduction, explain what the new device is going to address compared with traditional devices
- Research Gap needs to be merged with the introduction.
- In the introduction, the last paragraph clarifies what your novelty is in this work.
- You have almost 4 pages for the introduction; it is too long. Please merge the two sections and make it shorter.
- In line 157, please type the number of the table instead of a question mark.
- Table 2 hasn't been used anywhere.
- For Table 2, explain how you prioritized the alternatives.
- Figures 4 and 5 show the same information. Maybe you can combine or clarify why you needed to present in this way?
- In Figure 8, the labels of the images are not readable.
- Why didn't you utilize the color space for your purpose?
- It seems you used Canny edges, but you didn't explain why or for what.
- The desktop GUI for the computer vision system lacks clarity. Consider incorporating a legend or explanatory elements to interpret the color scheme.
- the results that were presented in section 4.1.1. to 4.1.4. needs to compare with existing UV-C and show how real-time makes the condition in the Champer different.
- Section 4.1.5 is mixed with the Methodology section. Some of the content should be explained earlier, specifically in Section 3.
- Additionally, the manuscript does not clarify the rationale for using computer vision. It’s important to explain why computer vision was chosen, and why machine learning methods were not considered. Instead, classical approaches were used, even though they did not yield satisfactory results and appear to be sensitive to environmental conditions.
- Furthermore, the manuscript does not explain the reason for the blurred images. If the blurring is due to contextual or environmental conditions, this should be clearly stated and discussed.
Author Response
Comment 1: The literature review has not adequately covered the novelty of this work.
Response 1: Thank you for your observation. In this revised version, the Introduction has been reorganized, and the literature review has been thoroughly updated to clearly present the novelty and contributions of this work, highlighting how it advances previous studies and addresses existing gaps in UV-C seed disinfection and vision-based monitoring systems.
Comment 2: In the introduction, explain what you applied and what was done on corn, including the use of a camera.
Response 2: Thank you for your comment. In response, the Introduction has been expanded to provide a clearer explanation of the applied methodology and what was done specifically with maize seeds, including the integration of a camera for monitoring. In particular, a new paragraph was added at line 69, presenting relevant studies on the application of computer vision in maize and broader agricultural contexts, detailing different methods and their practical applications. This addition highlights the rationale for using computer vision in the current study and situates our work within the existing literature.
Comment 3: Explain how innovative it is to use a camera for monitoring the color of corn in this study.
Response 3: Thank you for your comment on the novelty of using a camera to monitor maize seed color. In the revised manuscript, this innovative aspect is highlighted in line 96. Additionally, the Discussion section in line 587 emphasizes that the camera enables real-time, non-destructive monitoring of UV-C-induced color changes on individual seeds, offering a practical and efficient alternative to the indirect or labor-intensive methods previously reported in the literature.
Comment 4: In the introduction, explain what the new device is going to address compared with traditional devices
Response 4: Thank you for your comment. In the revised manuscript, the introduction has been improved to better contextualize the work. Paragraph 53 now highlights the features and limitations of existing prototypes, while line 83 clearly presents the approach proposed in this study. Additionally, the Methodology section (line 111) includes a design framework based on previous work, identifying the capabilities lacking in previously published prototypes and justifying the innovations introduced in our system.
Comment 5: Research Gap needs to be merged with the introduction
Response 5: Thank you for your suggestion. The manuscript has been reorganized so that the Research Gap is now integrated into the Introduction, improving the clarity and flow of the background and motivation for the study.
Comment 6: In the introduction, the last paragraph clarifies what your novelty is in this work.
Response 6: Thank you for your suggestion. The last paragraph of the Introduction has been revised to clearly highlight the novelty of this work, as recommended.
Comment 7: You have almost 4 pages for the introduction; it is too long. Please merge the two sections and make it shorter.
Response 7: The Introduction has been reorganized, and the previous separate sections have been merged, resulting in a more concise and better-structured presentation. This has shortened the text and improved its clarity and structure, providing a more concise presentation of the context, research gap, and novelty of this work. Thank you for your comment.
Comment 8: In line 157, please type the number of the table instead of a question mark.
Response 8: Thank you for the observation. The correction has been made, and the figures have been properly arranged in this revised version.
Comment 9: Table 2 hasn't been used anywhere. It was the correction of line 157
Response 9: Thank you for the observation. The tables have been properly arranged in this revised version.
Comment 10: For Table 2, explain how you prioritized the alternatives.
Response 10: As noted in section 2.3, alternatives were prioritized based on cost, development time, reliability, scalability, and regulatory compliance. Thus,the alternatives in Table 2 received the highest scores for convenience from the evaluation group, as stated in line 174. These selections reflect both technical feasibility and alignment with user needs and operational constraints, offering an optimal balance of performance, implementation effort, and regulatory compliance.
Comment 11: Figures 4 and 5 show the same information. Maybe you can combine or clarify why you needed to present in this way?
Response 11: Thank you for your comment. Figure 4 has been removed from the manuscript due to its redundancy and lack of added value.
Comment 12: In Figure 8, the labels of the images are not readable.
Response 12: Thank you for your observation. In the revised manuscript, the label size in Figure 8 has been increased to improve readability. Due to the reorganization of the paper, this figure is now presented as Figure 6.
Comment 13: Why didn't you utilize the color space for your purpose?
Response 13: Thank you for your comment. We are not completely certain about the exact intent of your question, but we assume it refers to the use of the RGB color space. As mentioned in the manuscript (paragraph starting in line 253), the RGB space is highly susceptible to illumination changes and device variations. In contrast, the CIELAB (LAB) color space is device-independent, perceptually uniform, and consistent across applications, making it more accurate and reliable for quantifying color differences in our seed analysis.
Comment 14: It seems you used Canny edges, but you didn't explain why or for what.
Response 14: Thank you for your comment. In the revised version, we clarified the rationale for using the Canny edge detection algorithm. In the proposed system, this step simplifies the image by enhancing edge information, making it easier to detect seed contours. However, it should be emphasized that the vision system was implemented as a proof-of-concept, intended primarily to demonstrate the feasibility of integrating computer vision with UV-C treatment systems. Future work will explore more advanced and robust methods to improve contour detection and analysis.
Comment 15: The desktop GUI for the computer vision system lacks clarity. Consider incorporating a legend or explanatory elements to interpret the color scheme.
Response 15: Thank you for your valuable suggestion. We agree that incorporating a legend or explanatory elements to clarify the color scheme would significantly improve the clarity of the desktop GUI. However, due to space constraints in this version, it was not possible to include these additions. It is important to note that the interface is still in an early stage of development, and your recommendation will be considered for future iterations to enhance usability and interpretation.
Comment 16: The results that were presented in section 4.1.1. to 4.1.4. needs to compare with existing UV-C and show how real-time makes the condition in the Champer different.
Response 16: Thank you for your observation. In the revised manuscript, we have emphasized how the integration of real-time sensor-based feedback allowed us to extract critical insights not possible with earlier prototypes, which lacked internal monitoring capabilities. As now presented in line 451 in the discussion section, the new prototype demonstrated several advancements:
-Lamp warm-up behavior was characterized, showing that stable UV-C output is achieved after approximately one minute—an aspect neglected in previous designs, potentially leading to inaccurate exposure times.
-Environmental monitoring revealed how temperature and humidity variations occur inside the closed chamber, and how the inclusion of an air extraction system improved stability, albeit with a minor reduction in UV-C intensity.
-Radiation mapping identified non-uniform distribution across the chamber floor, highlighting high- and low-radiation zones and underscoring the importance of seed placement for consistent treatment.
-Material attenuation analysis showed a measurable reduction in UV-C intensity through the cellophane seed holder, underlining the need to incorporate this factor into dose calibration to avoid underexposure.
Together, these insights illustrate how the proposed system advances beyond earlier prototypes by combining top–bottom lamp configurations with real-time monitoring of radiation, temperature, and humidity. This not only ensures reproducibility and safety but also provides actionable information for optimizing seed treatment—something that previous prototypes were unable to achieve.
Comment 17: Section 4.1.5 is mixed with the Methodology section. Some of the content should be explained earlier, specifically in Section 3.
Response 17: Thank you for this valuable comment. Following your suggestion, Section 4.1.5 has been reorganized, and part of its content has been moved to Section 2.4.4. (Methodology), where it is more appropriate. This adjustment not only clarified the methodological flow but also improved the overall organization and readability of the paper.
Comment 18: Additionally, the manuscript does not clarify the rationale for using computer vision. It’s important to explain why computer vision was chosen and why machine learning methods were not considered. Instead, classical approaches were used, even though they did not yield satisfactory results and appear to be sensitive to environmental conditions
Response 18: Thank you for this insightful comment. In the current work, the rationale for using computer vision lies in testing a proof-of-concept system that can operate without the need for training new datasets each time a modification is introduced, as would be required with machine learning approaches. This allowed us to rapidly evaluate feasibility and integration with the UV-C system. While we acknowledge that classical methods may show sensitivity to environmental conditions and do not always yield optimal results, they provided a suitable foundation for demonstrating the concept. For future iterations, we highly consider the use of machine learning techniques to improve robustness and adaptability. Additionally, as emphasized in the Discussion section (line 587), the use of a camera enables real-time, non-destructive monitoring of UV-C-induced color changes on individual seeds, offering a practical and efficient alternative to the indirect or labor-intensive methods previously reported in the literature. As a consequence, we are interested in developing further research in this line to create a more powerful and reliable prototype..
Comment 19: Furthermore, the manuscript does not explain the reason for the blurred images. If the blurring is due to contextual or environmental conditions, this should be clearly stated and discussed.
Response 19: Thank you for pointing this out. The blurred appearance of the images is not due to contextual or environmental conditions but rather results from the image-processing pipeline. Most edge detection algorithms, including the one applied in this work, employ Gaussian blurring as a pre-processing step to mitigate noise in the images. This reduces the likelihood of detecting spurious edges and allows for the identification of more precise and robust contours, improving the overall reliability of the edge detection process. Since the vision system is presented here as a proof-of-concept for integration with UV-C seed treatment, these preprocessing steps were intentionally adopted to demonstrate feasibility rather than to optimize final image quality.
Reviewer 4 Report
Comments and Suggestions for Authors
The following adjustments are needed before further consideration:
Do not use abbreviations in the abstract
try reducing the reliance on bullet points in your methodology and use a more straightforward descriptive writing style (Particularly Line 98-107 and 128-133)
The arrangement of the article is somewhat confusing. State of the art and research gap contains some of the methodology, also the results do (e.g: the beginning of Experimentation with corn seed samples to determine the benefits of the UV-C radiation process). Arrange the manuscript more traditionally so that each section clearly tackles its purpose.
Line 41-43: Before you state that “UV-C radiation exhibits potent germicidal effects, disrupting the DNA of pathogens …” you should state that “Similar to other bioactive radiation types, UV radiation is widely recognized in pest and pathogen control strategies” use a relevant reference to support this statement such as:
Nik Abdull Halim, N. M., Che Dom, N., Mat Seleei, N. M., Mohd Jamili, A. F., & Precha, N. (2025). Radiation effects on dengue vectors: A systematic review. DYSONA-Applied Science, 6(1), 160-171. https://doi.org/10.30493/das.2024.477813
Line 375-384: this is not a common or preferable academic writing style. It is better to describe the figures while showcasing your results instead of stating the figure name and its content.
In (Germination Test) I could not find the germination percentages. Did you only investigate fungal infection rates? If so, you should adjust 4.2.1. Germination Test (Infested Seed Count) appropriately.
Fig. 3 and 6 need enhancement
Some figures (e.g. Fig. 15 and 17) contain empty spaces. Try to reduce empty spaces in figures.
Add statistics on Fig. 17
The captions of figures and tables should be more informative
Enhance the discussion by adding limitation and connect your study with previous studies
Provide a detailed response to each of these comments and how you tackled them in the following revised draft.
Author Response
Comment 1: The following adjustments are needed before further consideration: Do not use abbreviations in the abstract. Try reducing the reliance on bullet points in your methodology and use a more straightforward descriptive writing style (Particularly Line 98-107 and 128-133).
Response 1: Thank you for pointing this out. In the revised manuscript, all abbreviations have been removed from the abstract to improve readability. Additionally, the methodology section has been restructured: the bullet points previously used in lines 98–107 and 128–133 were removed and rewritten in a straightforward descriptive style (line 117), as you suggested.
Comment 2: The arrangement of the article is somewhat confusing. State of the art and research gap contains some of the methodology, also the results do (e.g: the beginning of Experimentation with corn seed samples to determine the benefits of the UV-C radiation process). Arrange the manuscript more traditionally so that each section clearly tackles its purpose
Response 2: Thank you for this valuable suggestion. The manuscript has been reorganized to follow a more traditional structure, ensuring that each section clearly addresses its intended purpose. As a result, the clarity of the paper and the presentation of key concepts have been significantly improved. Specifically, part of Section 2 from the previous version has been integrated into the Introduction, while the remaining content was relocated to the Materials and Methods section as appropriate.
Comment 3: Line 41-43: Before you state that “UV-C radiation exhibits potent germicidal effects, disrupting the DNA of pathogens …” you should state that “Similar to other bioactive radiation types, UV radiation is widely recognized in pest and pathogen control strategies” use a relevant reference to support this statement such as:
Nik Abdull Halim, N. M., Che Dom, N., Mat Seleei, N. M., Mohd Jamili, A. F., & Precha, N. (2025). Radiation effects on dengue vectors: A systematic review. DYSONA-Applied Science, 6(1), 160-171. https://doi.org/10.30493/das.2024.477813
Response 3: Thank you for the suggestion. The statement you recommended was highly relevant to our research and has been incorporated into the manuscript, along with the suggested reference. These changes can be found in line 45 of the revised version.
Comment 4: Line 375-384: this is not a common or preferable academic writing style. It is better to describe the figures while showcasing your results instead of stating the figure name and its content.
Response 4: Thank you for your suggestion. The content from the mentioned lines has been revised and integrated using the academic style you recommended. The changes can be found in line 412 of the updated manuscript.
Comment 5: In (Germination Test) I could not find the germination percentages. Did you only investigate fungal infection rates? If so, you should adjust 4.2.1. Germination Test (Infested Seed Count) appropriately.
Response 5: Thank you for the observation. As the results presented focus exclusively on fungal infection rates, the section title has been adjusted accordingly in line 381 to reflect this specificity.
Comment 6: Fig. 3 and 6 need enhancement
Response 6: Thank you for your valuable observation. Both figures have been revised and enriched to enhance the clarity of the presented concepts. Due to the reorganization of the manuscript, they are now presented as Figure 1 and Figure 3 in the new version, instead of Figures 3 and 6.
Comment 7: Some figures (e.g. Fig. 15 and 17) contain empty spaces. Try to reduce empty spaces in figures.
Response 7: Thank you for your suggestion. The figures have been revised to eliminate empty spaces and now follow the format recommended in your comment.
Comment 8: Add statistics on Fig. 17
Response 8: Thank you for your valuable suggestion regarding the addition of statistics to Figure 17. In the revised manuscript, the figure has been updated to include the results of the Tukey test, which allows a more robust comparison between treatments. Due to the reorganization of the manuscript, this figure now appears as Figure 16.
Comment 9: The captions of figures and tables should be more informative
Response 9: Thank you for your valuable suggestion. The captions of all figures and tables have been revised to be more informative, providing clearer descriptions as recommended.
Comment 10: Enhance the discussion by adding limitation and connect your study with previous studies
Response 10: Thank you for your valuable comment. The Discussion section has been reorganized and expanded to clarify the limitations of the current study and to better connect our findings with previous research. As described in line 451, the new prototype demonstrated several advancements:
- Lamp warm-up behavior was characterized, showing that stable UV-C output is achieved after approximately one minute—an aspect neglected in previous designs, potentially leading to inaccurate exposure times.
- Environmental monitoring revealed how temperature and humidity variations occur inside the closed chamber, and how the inclusion of an air extraction system improved stability, albeit with a minor reduction in UV-C intensity.
- Radiation mapping identified non-uniform distribution across the chamber floor, highlighting high- and low-radiation zones and underscoring the importance of seed placement for consistent treatment.
- Material attenuation analysis showed a measurable reduction in UV-C intensity through the cellophane seed holder, underlining the need to incorporate this factor into dose calibration to avoid underexposure.
Together, these insights illustrate how the proposed system advances beyond earlier prototypes by combining top–bottom lamp configurations with real-time monitoring of radiation, temperature, and humidity. This not only ensures reproducibility and safety but also provides actionable information for optimizing seed treatment—capabilities that were absent in prior designs.

Round 2
Reviewer 3 Report
Comments and Suggestions for Authors
I don't have any more comments.
Reviewer 4 Report
Comments and Suggestions for Authors
Accepted in the present form